# Spatial Heterogeneity and Scale Effects of Transportation Carbon Emission-Influencing Factors—An Empirical Analysis Based on 286 Cities in China

**DOI:** 10.3390/ijerph20032307

**Published:** 2023-01-28

**Authors:** Tao Wang, Kai Zhang, Keliang Liu, Keke Ding, Wenwen Qin

**Affiliations:** 1Chongqing Transport Planning and Research Institute, Chongqing 401120, China; 2Shanxi Environmental Protection Institute of Transport, Taiyuan 030000, China; 3School of Traffic and Transportation, Chongqing Jiaotong University, Chongqing 400000, China; 4School of Economics and Business Administration, Chongqing University of Education, Chongqing 400000, China; 5Faculty of Traffic Engineering, Kunming University of Science and Technology, Kunming 650504, China

**Keywords:** transportation carbon emission, spatial autocorrelation, spatial heterogeneity, multiscale geographically weighted regression, China

## Abstract

In order to scientifically evaluate the characteristics and impact outcomes of transportation carbon emissions, this paper uses the panel statistics of 286 cities to measure transportation carbon emissions and analyze their spatial correlation characteristics. Afterwards, primarily based on the current research, a system of indicators for the impact factors of transportation carbon emissions was established. After that, ordinary least squares regression, geographically weighted regression, and multiscale geographically weighted regression models were used to evaluate and analyze the data, and the outcomes of the multiscale geographically weighted regression model were selected to analyze the spatial heterogeneity of the elements influencing transportation carbon emissions. The effects exhibit that: (1) The spatial characteristics of China’s transportation carbon emissions demonstrate that emissions are high in the east, low in the west, high in the north, and low in the south, with high-value areas concentrated in the central cities of Beijing-Tianjin-Hebei, the Yangtze River Delta, the Guangdong-Hong Kong-Macao region, and the Chengdu-Chongqing regions, and the low values concentrated in the Western Sichuan region, Yunnan, Guizhou, Qinghai, and Gansu. (2) The spatial heterogeneity of transportation carbon emissions is on the rise, but the patten of local agglomeration is obvious, showing a clear high-high clustering, and the spatial distribution of high-high agglomeration and low-low agglomeration is positively correlated, with high-high agglomeration concentrated in the eastern region and low-low agglomeration concentrated in the western region. (3) The effects of three variables—namely, GDP per capita, vehicle ownership, and road mileage—have a predominantly positive effect on transportation carbon emissions within the study area, while another three variables—namely, constant term, population density, and number of people employed in transportation industry—have different mechanisms of influence in different regions. Constant term, vehicle ownership, and road mileage have greater impacts on transportation carbon emissions.

## 1. Introduction

As a fundamental and pioneering industry in the development of various sectors and an important energy-consuming and carbon-emitting industry, the transportation industry is in an important period for achieving a transition to low-carbon options and fulfilling international carbon emission reduction commitments. Transportation carbon emissions involve many aspects of society, including the economy, technology, and policy and is an important research object for many subjects, such as the environment, transportation, the economy, etc. [1,2]. It is of great practical value to study how to effectively reduce the current carbon emission intensity of the transportation industry.

Currently, macroscopic studies related to transportation carbon emissions have focused on emission characteristics and driving factors. Studies have shown that transportation carbon emissions are mainly influenced by the following human factors: population [3,4,5,6], affluence [4,7,8,9,10,11], and the level of technology [12,13,14]. In terms of population, the vast majority of scholars have adopted population size as a quantitative indicator. Timilsna [3] analyzed the factors influencing carbon emissions from the transport sector in major Asian countries and concluded that GDP, population growth, and transport intensity are the most important influencing factors. Chen [5] analyzed transportation carbon emissions in Beijing and found that population size can influence transportation carbon emissions by affecting the intensity of transportation trips and energy use. On the economic side, all relevant studies have shown that economic development is the main influencing factor on transportation carbon emissions. Some scholars believe that economic development and industrial growth are the most important reasons for the growth of transportation carbon emissions. Zhu [4] found that economic growth is an important factor driving the growth of transportation carbon emissions through his study of the Beijing-Tianjin-Hebei region. In terms of science and technology, most of the existing studies have concluded that developments in science and technology will lead to a reduction in transportation carbon emissions. For instance, Xu [12,13] measured the influencing factors on transportation carbon emissions in different transportation development water regions and found that the degree of technological innovation impacts transportation carbon emissions, while energy use efficiency has a greater impact on transportation carbon emissions. Talbi [14] used a VAR model to analyze the influencing factors on transportation carbon emissions in Tunisia between 1980 and 2014 and discovered that better energy efficiency is the main driver for reducing transportation carbon emissions. In addition to the above three aspects, transportation carbon emissions are also influenced by other factors, such as the development of the transportation industry, and the human, natural, and social environment [15]. In summary, current studies on the influencing factors affecting transportation carbon emissions have focused on three aspects—namely, population, economic development, and technological development—and have mostly neglected comparative studies on the regional differences in the main drivers of transportation carbon emissions. Analysis of the spatial and temporal effects on transportation carbon emissions mostly focused on macroscopic areas, and most scholars analyzed them on national and provincial scales. However, due to the significant differences in the geographic locations, economic development levels, and resource endowments of different cities in China’s provincial areas, if research is limited to macroscopic levels, i.e., national and provincial areas, the relevant policies and recommendations proposed will lack relevance.

In summary, we have explored the influencing factors on transportation carbon emission differences and their spatial variation patterns using panel data of Chinese cities. Our research results can provide practical references for the differentiation of transportation carbon emission reduction policies in different regions, which is important for achieving emission reduction targets, accelerating supply-side reform, and speeding up the construction of an ecological civilization.

## 2. Materials and Methods

### 2.1. Methodology

#### 2.1.1. Transportation Carbon Emission Measurement

The existing literature summarizes two relatively well-established methods for calculating transportation carbon emissions: a top-down approach based on energy end-consumption [16,17] and a bottom-up approach based on data such as the number of vehicles, annual mileage, and energy consumption per mile [18,19,20]. The top-down approach, also known as the end-consumer-side approach, is a traditional carbon emission calculation method that uses a summary of the carbon emission factors of various energy consumption and corresponding energy sources in the study area. Based on a bottom-up algorithm, Schipper [21] proposed the concept of activity-to-mode ratio density fuel consumption, which multiplies the total energy consumption of various modes of transportation in a country or region by various energy CO_2_ emission factors to obtain carbon emissions. Due to the diversity of regions and the incompleteness of statistics in China, it is difficult to obtain the energy consumption of each urban transportation sector. Therefore, a “bottom-up” algorithm was used to calculate the transportation carbon emissions for each province and city in China, mainly including freight and passenger transportation. According to the literature, the microscopic model of freight transportation carbon emissions is as follows:(1)$$C_{f}=V_{j,k}\times CF_{j,k}\times r_{i},$$
where $C_{f}$ is freight transportation carbon emissions, j is transportation mod, k is means of transport, $V_{j,k}$ is freight turnover, $CF_{j,k}$ is unit energy consumption of different means of transport, and $r_{i}$ is carbon emission factor.

The passenger transportation carbon emission model is as follows:(2)$$C_{i}=S_{i}\times M_{i}\times N_{i,l}\times r_{l},$$
where $C_{i}$ is passenger transportation carbon emissions, $S_{i}$ is the number of motor vehicles, $M_{i}$ is driving distance, $N_{i,l}$ is energy consumption, and $r_{l}$ is carbon emission factor.

#### 2.1.2. Spatial Autocorrelation Characteristics of Transportation Carbon Emissions

Spatial autocorrelation of transportation carbon emissions refers to the potential dependency between the transportation carbon emissions of different cities and can be divided into global spatial autocorrelation and local spatial autocorrelation.

(1)Global spatial autocorrelation

The global Moran’s I index is generally used to reflect global spatial autocorrelation and takes on a value in the range of (−1, 1). Positive values indicate positive spatial correlation (high-high aggregation or low-low aggregation), negative values indicate negative spatial correlation (high-low aggregation), and a value of 0 indicates that the variables are independent within the study space. The global Moran’s I index is calculated using the following formula:(3)$$I=\frac{n\sum_{i=1}^{n}\sum_{j=1}^{n}w_{ij}\left(y_{i}-\bar{y}\right)\left(y_{j}-\bar{y}\right)}{\sum_{i=1}^{n}\sum_{j=1}^{n}w_{ij}\sum_{i=1}^{n}{\left(y_{i}-\bar{y}\right)}^{2}}=\frac{\sum_{i=1}^{n}\sum_{j=1}^{n}w_{ij}\left(y_{i}-\bar{y}\right)\left(y_{j}-\bar{y}\right)}{S^{2}\sum_{i=1}^{n}\sum_{j=1}^{n}w_{ij}},$$
where n is the number of space units, $y_{i}$ and $y_{j}$ are the observed values of spatial units, and $w_{ij}$ is the elements in the spatial weight matrix.

(2)Local spatial autocorrelation

Local Moran’s I index and LISA aggregation maps are used to measure and analyze the local spatial autocorrelation of transportation carbon emissions, and the local Moran’s I index is calculated as follows:(4)$$I_{i}=\frac{n\left(y_{i}-\bar{y}\right)\sum_{j=1}^{n}w_{ij}\left(y_{j}-\bar{y}\right)}{\sum_{i=1}^{n}{\left(y_{i}-\bar{y}\right)}^{2}}=\frac{\left(y_{i}-\bar{y}\right)\sum_{j=1}^{n}w_{ij}\left(y_{i}-\bar{y}\right)}{S^{2}},$$
where I is spatial correlation index,n, $y_{i}$, $y_{j}$, $w_{ij}$, and $S^{2}$ are the same as above. I>0 indicates that area i is positively correlated with its surrounding area, which indicates high-high agglomeration or low-low agglomeration. I<0 indicates the opposite.

#### 2.1.3. Analysis of the Influencing Factors of Transportation Carbon Emissions

As different regions have different development characteristics of transportation carbon emissions, this paper uses Geographically Weighted Regression (GWR) models and Multiscale Geographically Weighted Regression (MGWR) models, which are improved on the basis of GWR models, to compare with Ordinary Least Squares (OLS) models. By comparing the fit and explanatory power of the two types of models, the spatial variation of the factors affecting transportation carbon emissions are explored.

(1)Ordinary Least Squares

Ordinary least squares regression is a common linear regression and global estimation model which uses the following equation:(5)$$Y_{i}=\beta_{0}+\beta_{1}x_{i1}+\beta_{2}x_{i2}+\cdots +\beta_{m}x_{im}+\varepsilon_{i},$$
where $Y_{i}$ is transportation carbon emissions, $\beta_{0}$ is constant term, $x_{im}$ is influencing factor, $\beta_{m}$ is regression coefficient, and $\varepsilon_{i}$ is random error. In addition, the OLS model does not require the distribution of the independent variable, but the random error term needs to conform to a normal distribution with a mean of 0. Therefore, the Kolmogorov–Smirnov test (K–S test) needs to be performed on the dependent variable of the OLS model, and if the test is not satisfied, the Box-Cox transformation is performed so that the dependent variable after the transformation is linearly correlated with the independent variable and the random error term obeys a normal distribution. The transformation is performed as follows:(6)$$Y_{i}(\gamma )=\{\begin{array}{c} \frac{Y_{i}^{\gamma }-1}{\gamma },\gamma \neq 0\\ \ln Y_{i},\gamma =0 \end{array},$$
where $Y_{i}(\gamma )$ is variable after conversion and γ is conversion parameters. Maximum likelihood estimation and Bayes method can be used for estimation.

(2)Geographically Weighted Regression

The geographically weighted regression model is an improved model based on a linear regression model with the addition of parameters reflecting differences in geographic location to differentiate the variables at a local scale. Its regression equation is as follows:(7)$$Y_{i}=\beta_{0}\left(u_{i},v_{i}\right)+\sum_{m}\beta_{m}\left(u_{i},v_{i}\right)x_{im}+\varepsilon_{i},$$
where $Y_{i}$ is transportation carbon emissions, $\left(u_{i},v_{i}\right)$ are longitude and latitude, $\beta_{0}\left(u_{i},v_{i}\right)$ is constant term, $\beta_{m}\left(u_{i},v_{i}\right)$ is regression coefficient; $x_{im}$ is explanatory variables, and $\varepsilon_{i}$ is random error. The $\beta_{m}\left(u_{i},v_{i}\right)$ is a distance attenuation function; the weighted least squares method is used for the calculation, which is calculated as follows:(8)$$\beta_{m}\left(u_{i},v_{i}\right)={\left[X^{T}W\left(u_{i},v_{i}\right)X\right]}^{-1}X^{T}W\left(u_{i},v_{i}\right)Y,$$
where X is independent variable, Y is dependent variable, and $W\left(u_{i},v_{i}\right)$ is spatial weight matrix. When determining the bandwidth, the minimum information criterion method (AICc) is generally chosen, and its calculation formula is as follows:(9)$$AICc=\frac{2t-2\ln\left[L({\hat{\theta }}_{i},x)\right]}{i},$$
where t is the number of independent parameters of GWR, ${\hat{\theta }}_{i}$ is the maximum likelihood estimate, and $L({\hat{\theta }}_{i},x)$ is the maximum likelihood function of θ.

(3)Multiscale Geographically Weighted Regression

The multiscale geographically weighted regression model finds the most appropriate bandwidth for each variable through stepwise regression, thus better demonstrating the scale of action and spatial characteristics of each variable [22,23,24,25,26,27]. The regression equation of the multiscale geographically weighted regression model is as follows:(10)$$Y_{i}=\sum_{m=1}^{k}\beta_{bwm}(u_{i},v_{i})x_{im}+\varepsilon_{i},$$
where $\beta_{bwm}$ is bandwidth, and the rest of the variables have the same meaning as above.

Unlike the geographically weighted regression model, all regression parameters of the multiscale geographically weighted regression model are solved by local regression, which accounts for the different bandwidths of different variables. In the solution process, multiscale geographically weighted regression still uses the bandwidth selection and kernel function selection rules of the geo-weighted regression. In most cases, the AICc criterion is chosen for the general bandwidth, while the quadratic kernel function is chosen for the kernel function. In addition, the solution process for geographically weighted regression is a weighted least squares estimation, while multiscale geographically weighted regression is similar to a generalized additive model with the following expressions:(11)$$Y=\sum_{m=1}^{k}f_{i}+\varepsilon ,$$
(12)$$f_{i}=\beta_{bwm}x_{m},$$

We chose to use geographically weighted regression as the preliminary estimate for a multiscale geographically weighted regression, and then solved for the difference between the predicted and true values to obtain the initialized residuals ε.
(13)$$\varepsilon =Y-\sum_{m=1}^{k}f_{i},$$

The initialized residuals are added to $f_{1}$ with the first independent variable $x_{1}$ for the geographically weighted regression operation to obtain the optimal bandwidth of $x_{1}$ and the new estimated parameters, then the corresponding estimated parameters in the multiscale geographically weighted regression model are updated and the new residual values are solved for, which are added to $f_{2}$. The above steps are then repeated until the regression coefficients and optimal bandwidths of all variables are completed, and then it is judged whether the convergence criterion is satisfied, which is generally determined by the proportional change in the sum of squared residuals, calculated as follows:(14)$$SOC_{RSS}=\left|\frac{RSS_{new}-RSS_{old}}{RSS_{new}}\right|,$$
where $SOC_{RSS}$ is proportion of change in the sum of squared residuals, $RSS_{new}$ is the residual sum of squares for this calculation, and $RSS_{old}$ is the residual sum of squares for last calculation.

### 2.2. Data

In the measurement of transportation carbon emissions, data on the holdings of different means of transportation were obtained from the statistical yearbooks of each city in previous years, the China Automobile Industry Yearbook, the China Economic and Social Development Statistical Database, and Social Development Statistical Bulletin. Carbon emission coefficients and the average fuel consumption of various energy sources were obtained from IPCC 2006 and the Guidelines for the Preparation of Provincial Greenhouse Gas Inventory. Mileage data were obtained from the China Urban Statistical Yearbook and the annual reports of urban transportation development in previous years. The energy consumption per unit of each transportation means is shown in Table 1. Data on administrative areas were obtained from the Data Center for Resource and Environmental Sciences of the Chinese Academy of Sciences, and data related to the influence factors of transportation carbon emissions were obtained from the statistical yearbook of each city.

## 3. Results

### 3.1. Transportation Carbon Emission Measurement Results

According to the measurement method detailed in Section 2.1.1, the carbon emission data of 287 cities collected between 2009 and 2019 were measured and visualized using spatial geographic information technology. As the research data showed a more regular linear change trend within the research period, the first, last, and the middle years generally tend to be selected as the characteristic years to briefly present their data change characteristics. According to this principle, 2009, 2014, and 2019 were selected as representative years for display, as shown in Figure 1.

From the spatial distribution of transportation carbon emissions in the three characteristic years, the spatial characteristics of China’s transportation carbon emissions show that emissions are high in the east, low in the west, high in the north, and low in the south, and these characteristics are stable. In terms of regional characteristics, the areas with high-value transportation carbon emissions are mainly concentrated in the central cities of the Beijing-Tianjin-Hebei region, the Yangtze River Delta, the Guangdong-Hong Kong-Macao region, and the Chengdu-Chongqing regions, and the low values are mainly concentrated in the Western Sichuan region, Yunnan, Guizhou, Qinghai, Gansu, and other regions. In terms of city characteristics, Shanghai, Chongqing, Tianjin, Suzhou, Shijiazhuang, Tangshan, Wuhan, Chengdu, and Ordos are stably ranked in the top ten with regard to emissions. From the temporal changes in spatial distribution, the transportation carbon emissions in Eastern Inner Mongolia and Central Xinjiang show faster growth, and the transportation carbon emissions in the Beijing-Tianjin-Hebei region have a trend of regional spread growth.

### 3.2. Spatial Autocorrelation Characteristics of Transportation Carbon Emissions

#### 3.2.1. Global Spatial Autocorrelation

The Moran’s I index was used to examine the transportation carbon emissions from the period 2009–2019, and Moran’s scatter plots were plotted, producing the results shown in Figure 2. The horizontal coordinate represents the deviation of the observed value from the mean, and the vertical coordinate indicates the spatial lag value. Each quadrant corresponds to a different type of spatial autocorrelation: the first quadrant (H-H) indicates high-high positive correlation, the second quadrant (L-H) indicates low-high negative correlation The third quadrant (L-L) indicates a low-low positive correlation, and the fourth quadrant (H-L) indicates a high-low negative correlation.

According to the test results, the global Moran’s I index values for each year were greater than 0. Combined with the *p*-values, it can be seen that at a significance level of 0.1, transportation carbon emissions from the period 2009–2019 show a more significant positive spatial correlation, i.e., areas with higher transportation carbon emissions tend to have one or more neighboring cities that also have higher carbon emissions (high-high positive correlation), while cities with low transportation carbon emissions also have at least one city with low transportation carbon emissions adjacent to them (low-low positive correlation). The global Moran’s I index value has remained stable at around 0.3 for 11 years, indicating a long-term stable, positive spatial correlation for transportation carbon emissions at a municipal scale in China. The number of samples located in the L-H and H-L quadrants accounted for 14%, 16%, and 19% of the total in 2009, 2014, and 2019, respectively. These results indicate that the spatial heterogeneity (discrete distribution pattern) of transportation carbon emissions showed an increasing trend during the study period, but in general, the spatial correlation of transportation carbon emissions at the local scale was high, and the local clustering pattern was significant.

#### 3.2.2. Local Spatial Autocorrelation Analysis

A characteristic annual transportation carbon emission LISA map was drawn based on the results of local Moran’s I calculation, and the results are shown in Figure 3. Overall, the L-L aggregation regions of China’s transportation carbon emissions are mainly distributed across Western China; the H-H aggregation regions are mainly distributed across Central China; the H-L aggregation regions are mainly distributed across Southwest China; and the L-H aggregation regions are mainly located near the H-H aggregation region.

As can be seen in Figure 3, the spatial characteristics have a certain stability in the characteristic years. Specifically, Xinjiang, Qinghai, Western and Southern Sichuan, Western Yunnan, and Hainan are L-L agglomeration areas, which may be due to the low level of overall socio-economic development and insufficient transportation infrastructure in the region. H-H agglomeration areas are mainly located in some of the cities in the Beijing-Tianjin-Hebei region, and some of the cities in Yangtze River Delta, which to a certain extent reflects the close economic contacts between cities within the two city groups. H-L agglomeration areas are mainly distributed across the Chengdu-Chongqing urban agglomeration, which is also consistent with the large differences in economic development between cities within the Chengdu-Chongqing urban agglomeration. The GDPs of Chongqing and Chengdu in 2020 were about 2.5 trillion and 1.77 trillion, respectively, placing them 5th and 7th in the national GDP city ranking, while Mianyang, the city ranked second with regard to GDP within the city cluster, had a GDP of slightly over 300 billion in 2020 and was then ranked 91st among the top 100 cities in China with regard to GDP. The huge developmental differences between the two major central and regional cities have led to an unbalanced distribution of various factors of production, as reflected in the difference in transportation carbon emissions, while transportation carbon emissions also reflect the development level of the cities to a certain extent.

### 3.3. Factors Influencing Transportation Carbon Emissions

#### 3.3.1. Indicator System

The index system of transportation carbon emission influence factors was mainly constructed using four elements: population, economy, science and technology, and transportation industry development. In terms of population, population density and the proportion of urban population were selected as indicators. In terms of economy, the following three indicators were selected: GDP per capita, the proportion of added value of tertiary industry, and GDP growth rate. In terms of science and technology, indicators related to energy use, such as energy technology and transportation energy intensity, were selected. In terms of transportation industry development, indicators such as road mileage, vehicle ownership, passenger transportation structure, freight transportation structure, and the number of employees in the transportation industry were selected [4,28,29,30,31]. The definition and calculation method of each indicator are detailed in Table 2. All data in the table were obtained from the statistical yearbooks of each city.

#### 3.3.2. Model Comparison

Based on the results of the transportation carbon emission measurements and the index system of transportation carbon emission influencing factors, the ordinary least squares regression model, geographically weighted regression model, and multiscale geographically weighted regression model were used to fit. We selected four indicators—namely, goodness-of-fit R^2^, adjusted R^2^, sum of squares of residuals, and AICc—to compare and analyze the fitting effects of ordinary least squares regression, geographically weighted regression, and multiscale geographically weighted regression models (Table 3). The larger the values of goodness-of-fit R^2^ and adjusted R^2^, and the smaller the values of AICc and sum of squares of residuals, the higher the model fitting accuracy.

As can be seen in Table 3, the R^2^ and adjusted R^2^ of the geographically weighted regression and multiscale geographically weighted regression models are remarkably higher than those of the ordinary least squares regression. The AICc and the sum of squared residuals of the multiscale geographically weighted regression are remarkably lower than those of the ordinary least squares regression and geographically weighted regression. Compared with ordinary least squares regression, the R^2^ and adjusted R^2^ of the multiscale geographically weighted regression model are higher by 0.28 and 0.22, respectively, on average. Compared with geographically weighted regression, the AICc and the sum of squared residuals of the multiscale geographically weighted regression model are lower by 313.23 and 53.00, respectively, on average. It can be determined that the results of the multiscale geographically weighted regression are better than the ordinary least squares regression and the geographically weighted regression, so we chose the model results of the multiscale geographically weighted regression for further analysis.

#### 3.3.3. Scale Analysis

Since multiscale geographically weighted regression can produce the optimal bandwidth (AICc) of each variable to reflect the differences between the action scales of different variables, the optimal bandwidth of each variable is the optimal action scale of each variable, and the AICc values of each variable are shown in Table 4.

Table 4 shows that there are large differences between the scales of action among the variables. In the model results of multiscale geographically weighted regression, intercept is the constant term of the regression model, indicating the effect of different spatial locations or locational conditions on transportation carbon emissions, with other variables unchanged. Since we controlled for demographic, economic, technological, and transportation factors, the realistic effect of locational conditions may include factors such as natural and social environment. The AICc values of the constant term are 60, 43, and 43 in 2009, 2014, and 2019, respectively, which are relatively low compared with other variables, and the scale is close to the provincial scale on average. This indicates that the effect of spatial location on transportation carbon emissions is essentially the same at the provincial scale. Moreover, the effect scale of NC is also low, at 43 for all years except for 2009, which indicates that there is a large spatial variation in transportation carbon emissions with changes in car ownership. The AICc values of the NTP are 137, 137, and 146 in 2009, 2014, and 2019, respectively, which are close to the regional or district scales on average. The AICc values of Agdp, FS, TEI, and PTS were all greater than 200 in most cases, which are close to the global scale, indicating that there is little spatial heterogeneity in the effects of the above factors on transportation carbon emissions. The five indicators of PD, CR, SR, ROAD, and ET are at the global scale in one year, which means that there is no spatial heterogeneity, and the above five indicators might be at the provincial scale or regional scale in the remaining years, as the spatial heterogeneity varies with the year.

#### 3.3.4. Spatial Distribution Characteristics of Coefficients

The *p* < 0.05 test was performed on all variables with the results of the model. According to the test results, the variables with a small number of test sample points were screened out, and the variables with a larger number of test sample points were further analyzed. Thus, we finally selected Intercept, PD, Agdp, NC, ROAD, and NTP for further analysis:(1)The spatial distribution of the constant terms was visualized, and the resulting images are shown in Figure 4. It can be seen in Figure 4 that the value range of the constant term was −0.39–0.70, indicating that the influence of location conditions on transportation carbon emissions has different influence mechanisms in different regions. The impact of location conditions on transportation carbon emissions shows an increasing trend over time. The insignificant area has gradually narrowed since 2009, from most of the eastern, southwestern, and northwestern regions to a small part of the northern region, indicating that the impact of location conditions on transportation carbon emissions is gradually expanding. In addition, there are obvious spatial differences in the impact of location conditions on transportation carbon emissions. Overall, the impact of location conditions on transportation carbon emissions is positive in Northeast and North China, and negative in East, Central, and South China. From the absolute value of the coefficient, the intensities of positive impact and negative impact are similar. However, there is a trend that the positive influence is gradually stronger than the negative influence in time.

(2)The spatial distribution of the coefficients of population density indicators was visualized, and the resulting images are shown in Figure 5. As can be seen in Figure 5, the coefficients of the population density index range from −0.28 to 0.68. Similar to the Intercept, the influence of population density on transportation carbon emissions has different influence mechanisms in different regions. In terms of the absolute value of the coefficient, the impact of population density on transportation carbon emissions has gone through a process of changing from a positive to a negative influence, then back to a positive influence. Contrary to geographic conditions, the impact of population density on transportation carbon emissions indicates a diminishing tendency over time. Starting in 2009, the relevant area gradually shrank from the entire region to a portion of Central, North, and Northeast China. The majority of South and Southwest China’s regions, which showed a negative impact in 2009, gradually transitioned to no discernible impact. Additionally, a significant spatially divergent characteristic may be seen in the relationship between population density and transportation carbon emissions. Population density has a substantial positive impact on transportation carbon emissions in Northeast China, a modest positive impact in North and East China, and a negative effect in the majority of the remaining regions in 2009. With the gradual reduction in the scope of the effect of population density on transportation carbon emissions, some prefecture-level cities in the northeast region changed from the original positive impact to a negative one by 2019. It is worth noting that the impact of population density on transportation carbon emissions in the Yangtze River Delta region has been showing a positive impact, and the intensity of the impact has been increasing.

(3)The results of visualizing the spatial distribution of GDP per capita indicator coefficients are displayed in Figure 6. As seen in Figure 6, the coefficients of GDP per capita indicators range from 0.08 to 0.27, indicating that in the study area, GDP per capita has a positive impact on transportation carbon emissions, reflecting the strong interaction between the economy and transportation carbon emissions in these cities. According to the coefficient values, the effect of GDP per capita on transportation emissions is weak and has become weaker over time. Overall, the relationship between GDP per capita and transportation emissions exhibits a clearer hierarchy from east to west and was constant over the course of the study. From the northwest region, the high-value region gradually spreads to the southwest and northeast regions as time goes on. The influence of GDP per capita on transportation carbon emissions was throughout the study period, indicating that GDP per capita is an important influencing factor on transportation carbon emissions and the influence mechanism is more certain. It is noteworthy that the effect of GDP per capita on transportation emissions gradually shifts from being positive between 2009 and 2014, to being negligible in several coastal areas of the southeast.

(4)The spatial distribution of road mileage indicator coefficients is visualized, and the results are shown in Figure 7. From Figure 7, it can be seen that the values of road mileage indicator coefficient range from 0.09 to 0.80, indicating that road mileage has a positive impact on transportation carbon emissions. Except for a few regions where there was no significant effect in 2009, road mileage had a significant effect on transportation carbon emissions across the whole study area in 2014 and 2019, which indicates that the influence of road mileage on transportation carbon emission is gradually expanding, and road mileage is an important indicator affecting transportation carbon emissions. In terms of the absolute value of the coefficients, the overall degree of influence of road mileage on transportation emissions is strong. Across time, the degree of influence of road mileage on transportation carbon emissions has tended to weaken, but is generally more stable, which may be due to the transformation of transportation structures, with the proportion of road transport to railway transport and water transport increasing year by year. In 2009, the degree of influence of road mileage on transportation carbon emissions was higher in North China, and then the region with higher influence gradually shifted to Northeast China. The degree of influence in North and Northwest China gradually weakened, and the low-value region showed an expansion, while other regions showed a more stable state across time.

(5)The spatial distribution of car ownership index coefficients was visualized, and the resulting images are shown in Figure 8. In Figure 8, it can be seen that the values of the car ownership index coefficients are all positive, indicating that the impact of car ownership on transportation carbon emissions is positive in the study area. Car ownership has a significant global impact on transportation carbon emissions in the selected characteristic years, suggesting that car ownership is an important influencing factor on transportation carbon emissions and that the mechanism of its influence is more affirmative. In terms of the values of the coefficients, similar to the road mileage indicator, car ownership has a great overall influence on transportation emissions. Similar to the GDP per capita indicator, the spatial distribution of car ownership coefficients has a clear hierarchical structure, showing a pattern of increasing from west to east and from south to north. The high-value area gradually spread from north to south over the time period, and the low-value area narrows, first covering most of Southwest and South China in 2009, and later a small part of Southwest China and part of South China in 2019, indicating that the degree of influence of car ownership on transportation carbon emissions tends to increase over the time period, and this trend is particularly evident in Southwest and Northwest China. This reflects the greater intensity in the use of small cars in these cities, indirectly indicating the lack of attractiveness of urban public transportation, which leads to an increase in the impact of car ownership on transportation carbon emissions.

(6)The spatial distribution of the index coefficient of the number of employees in the transportation industry was visualized, and the resulting images are shown in Figure 9. As can be seen in Figure 9, the coefficients of the number of employees in the transportation industry range from −0.94 to 0.21. Similar to the constant term and population density indicators, the impact of the number of people working in the transportation industry on transportation carbon emissions has different mechanisms in different regions. This may, in part, reflect a shift in the industry’s work patterns in some regions, such as a reduction in the number of employees and improved efficiency due to the smart developments in the industry. The number of employees in the transportation industry has no significant effect on transportation carbon emissions in most regions in the south, and in the regions where there is a significant effect, it is mainly a negative effect. In 2009, 2014, and 2019, the positive impact areas included a small part of Central China, a small part of Eastern China, and other areas scattered around the area. In terms of the absolute value of the coefficients, the degree of influence of the number of employees in the transportation industry on transportation carbon emissions shows a trend which transitions from weak to strong, and then back to weak. It is noteworthy that the influence of Kunming in the northwest and southwest has gradually changed from no significant impact in 2009 to a weak negative impact.

## 4. Discussion

This paper extends the macro-level study of transportation carbon emissions from national and provincial scales to city scales, and on this basis, the conclusions regarding the spatial and temporal variation characteristics of transportation carbon emissions are more reasonable. In addition, this paper used a multiscale geographically weighted regression model to analyze the influencing factors of transportation carbon emissions, which can accurately calculate the scale of action of each influencing factor, and the carbon emission reduction strategy of transportation industry based on this can be more targeted. Nevertheless, due to the limited data and research conditions, we selected the influencing factors of transportation carbon emissions from four dimensions: population, economy, technology, and transportation industry development, and the indicators were selected macroscopically, lacking further analysis of microscopic deep-level factors, such as individual awareness of low-carbon travel, and consideration of the influence of urban spatial structure and urban public transportation system on urban transportation carbon emissions. In addition, although multiscale geographically weighted regression makes up for the shortcomings of the traditional global regression model, it still has a strong linearity assumption. With the increasing maturity of machine learning and deep learning methods, the linearity assumption of the model should be further relaxed in the future, while exploring the impact of interaction effects among variables on transportation carbon emissions.

## 5. Conclusions

We measured the transportation carbon emissions based on the panel data of 286 cities in China, analyzed the spatial autocorrelation characteristics of transportation carbon emissions on the basis of the spatial interaction theory, and analyzed the spatial heterogeneity characteristics of transportation carbon emission influencing factors using the MGWR model considering multiscale effects. Based on this analysis, we came to the following conclusions:(1)China’s transportation carbon emissions showed the spatial characteristics of high emissions in the east, low emissions in the west, high emissions in the north, and low emissions in the south, and these characteristics were stable throughout the time period studied. At a local scale, transportation carbon emissions have a high spatial correlation and a clear pattern of local aggregation. The L-L agglomeration area is mainly due to the low level of overall regional socio-economic development and insufficient transportation infrastructure. Additionally, the H-H agglomeration area has close economic interactions within the area; the economic development of cities within the H-L agglomeration area varies greatly. We should focus on promoting transportation carbon emission reduction in H-H agglomeration regions, such as some cities in the Beijing-Tianjin-Hebei region and some cities in the Yangtze River Delta, and thus effectively reduce total transportation carbon emissions. As the central cities are closely linked to the surrounding, non-central cities, policies should be jointly formulated for such regions to promote regional transportation emission reduction linkage: for example, by designing a cross-regional, low-carbon, multimodal transportation system to cater for the interaction between regional human and logistical flows.(2)Location, population density, GDP per capita, car ownership, road mileage, and the number of employees in the transportation industry are the major factors influencing transportation carbon emissions. The impact of car ownership is low, which indicates that the spatial variation of transportation carbon emissions with car ownership is significant. The impact of the number of employees in the transportation industry is similar to that of the regional or district scale. The impacts of the other variables were similar to those on the global scale in a certain characteristic year, which indicates that there is little spatial heterogeneity in the effects of the above factors on transportation carbon emissions. Based on the differences in coefficients, differences in the growth mechanisms of carbon emissions in different cities can be identified, and thus targeted measures can be proposed. For example, the GDP per capita in Northwest China showed a stronger positive influence on transportation carbon emissions, reflecting the stronger coupling between urban economic development and transportation in Northwest China, which indicates that there should be a focus on promoting the low-carbon transformation of transportation in this region. Additionally, the degree of influence of car ownership on transportation carbon emissions showed an enhanced trend over the time period, and this trend was especially obvious in Southwest and Northwest China, reflecting the more intensive use of small cars in these cities. Indirectly, this indicates that urban public transportation is not attractive enough, so the development of public transportation should be increased to promote the green transformation of urban transportation.

## Figures and Tables

**Figure 1 ijerph-20-02307-f001:**
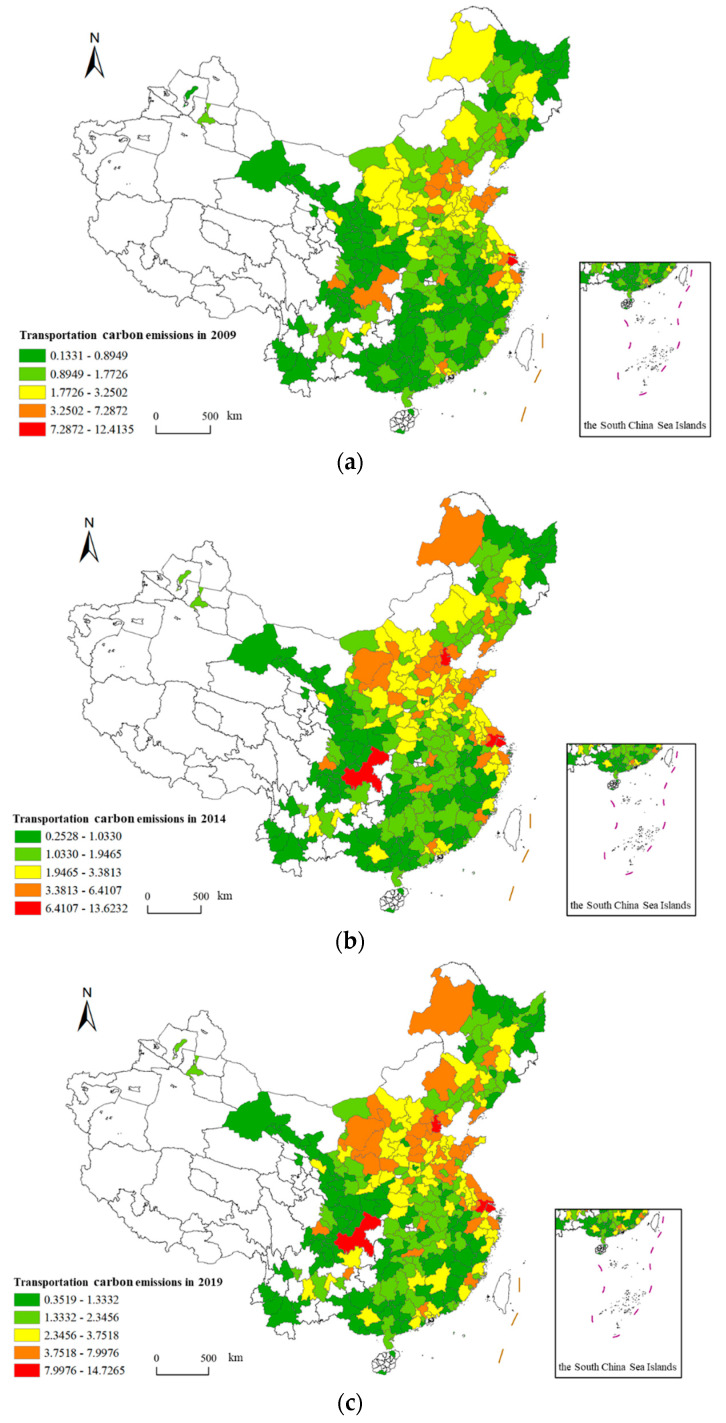
Spatial distribution of transportation carbon emissions in characteristic years. (**a**) Spatial Distribution of Transportation Carbon Emissions in 2009; (**b**) Spatial Distribution of Transportation Carbon Emissions in 2014; (**c**) Spatial Distribution of Transportation Carbon Emissions in 2019.

**Figure 2 ijerph-20-02307-f002:**
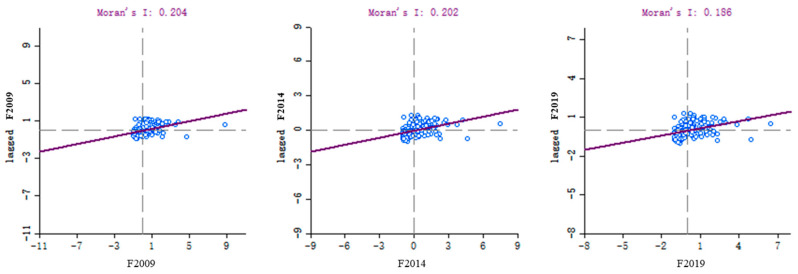
Moran scatter plots of characteristic years.

**Figure 3 ijerph-20-02307-f003:**
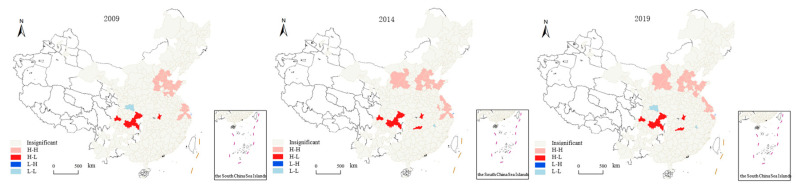
LISA aggregation map of transportation carbon emissions in characteristic years.

**Figure 4 ijerph-20-02307-f004:**
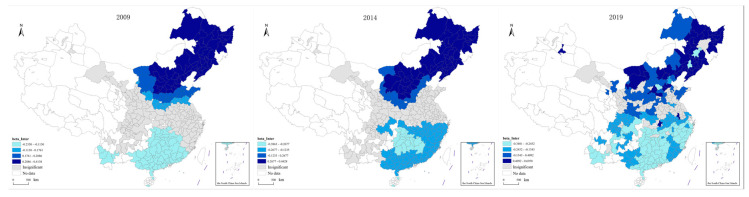
Spatial distribution of constant term.

**Figure 5 ijerph-20-02307-f005:**
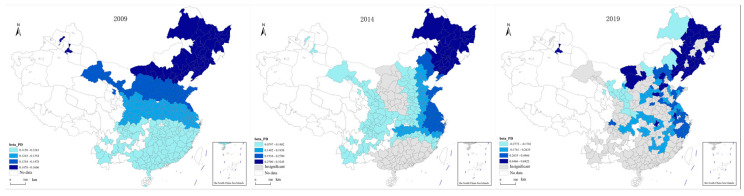
Spatial distribution of population density indicator coefficient.

**Figure 6 ijerph-20-02307-f006:**
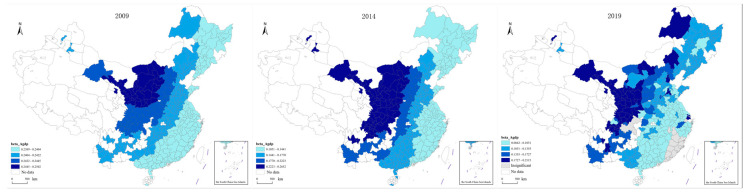
Spatial distribution of GDP per capita indicator coefficient.

**Figure 7 ijerph-20-02307-f007:**
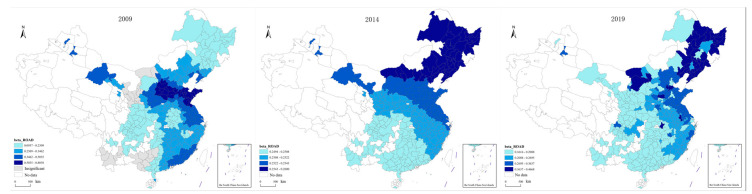
Spatial distribution of road mileage indicator coefficient.

**Figure 8 ijerph-20-02307-f008:**
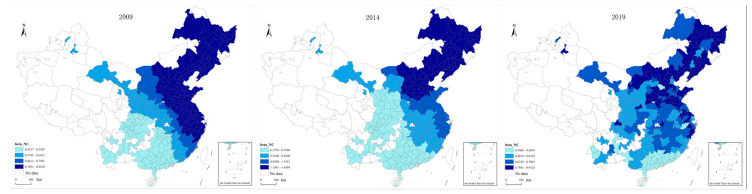
Spatial distribution of car ownership indicator coefficient.

**Figure 9 ijerph-20-02307-f009:**
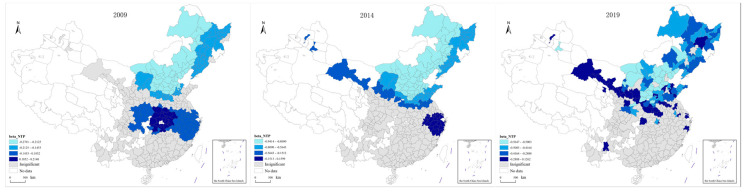
Spatial distribution of number of employees in the transportation industry indicator coefficient.

**Table 1 ijerph-20-02307-t001:** Energy consumption per unit of each mode of transport.

Means of Communications		Unit Energy Consumption
Road transport	Gasoline automobile	0.076 L
	Diesel automobile	0.060 L
Railway transportation	Electric locomotive	107.9 kwh
	Diesel locomotive	26.4 kg
	Steam locomotive	330.5 kg
Air transport	Aircraft	0.284 kg
Urban traffic	Rail transit	240 kwh
	Bus	40 L
	Private car	9 L
	Motorcycle	2 L
	Taxi	9 L

Note: In the calculation process, urban transportation is taken to be powered by fuel, except for rail transportation, which is powered by electricity.

**Table 2 ijerph-20-02307-t002:** Transportation carbon emission impact factor index system.

Variable		Symbol	Definition
Population size	Population density	PD	Ratio of resident population to area of jurisdiction
	Proportion of urban population	CR	Ratio of urban population to total population
Economic level	GDP per capital	Agdp	/
	Proportion of added value of tertiary industry	SR	Ratio of added value of tertiary industry to added value of all industries
	GDP growth rate	GDPR	/
Development of science and technology	Energy technology	ET	Ratio of gross regional product to total urban energy consumption
	Traffic energy intensity	TEI	Ratio of passenger and freight turnover to total energy consumption
Development of transportation industry	Road mileage	ROAD	/
	Passenger transport structure	PTS	Ratio of road passenger turnover to total passenger turnover
	Freight structure	FS	Ratio of road freight turnover to total freight turnover
	Car ownership	NC	/
	Number of employees in the transport industry	NTP	/

**Table 3 ijerph-20-02307-t003:** Model metrics of OLS, GWR, and MGWR.

Evaluating Indicator	OLS			GWR			MGWR		
	2009	2014	2019	2009	2014	2019	2009	2014	2019
R^2^	0.64	0.68	0.65	0.91	0.91	0.90	0.94	0.94	0.94
Adjusted R^2^	0.71	0.67	0.63	0.84	0.88	0.86	0.91	0.91	0.91
AICc	461.52	510.74	542.96	408.04	551.29	717.65	252.36	230.56	257.37
Sum of squares of residuals	71.77	90.28	100.81	44.56	60.04	109.23	18.73	18.28	17.82

**Table 4 ijerph-20-02307-t004:** AICc value of each variable.

Variable	AICc		
	2009	2014	2019
Intercept	60	43	43
PD	265	114	43
CR	213	50	50
Agdp	286	187	215
SR	41	116	285
GDPR	102	105	43
ROAD	44	284	154
FS	43	238	285
NC	153	43	43
TEI	286	207	124
PTS	286	284	250
ET	286	69	113
NTP	137	137	146

## Data Availability

The data showed in this study are available on request from the corresponding author.

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
