# Peer review of "Spatial Heterogeneity and Scale Effects of Transportation Carbon Emission-Influencing Factors—An Empirical Analysis Based on 286 Cities in China"

_ijerph, 2023, doi:10.3390/ijerph20032307_

Round 1
Reviewer 1 Report
This is an interesting paper that explores the factors influencing the differences in transportation carbon emissions and their spatial variation patterns. These data, by themselves, are valuable for making transportation carbon emission reduction policies in different regions.
In this paper, the transportation carbon emissions are determined by "bottom-up" algorithm method. I am wandering about the uncertainty of these results? Has the author compared the calculated results with a real value for some city that has the entire suite of statistical data?
Author Response
Thank you very much for reading this paper in your busy schedule and for your constructive comments and suggestions on the paper, which have greatly helped us to further improve the paper. We have carefully considered and analyzed your comments, and have revised the paper according to your opinions, and the revision is described as follows.
We also considered whether the transportation carbon emissions calculated in this paper are close to the real ones. Since the transportation carbon emission data for Chinese cities are not published but the transportation carbon emission data for each province are published. To address our doubts, we selected some provinces and compared our calculated transportation carbon emissions for each city in that province with the officially published transportation carbon emissions for that province after adding them together. We found that in most of the provinces, our calculations are close to the official figures, so we believed that our transportation carbon emission calculations are accurate.
In addition, we have edited the article extensively for English language and style, so please review it again.

Reviewer 2 Report
This paper evaluated the characteristics of transportation carbon emissions and numerical models were considered to analyze the spatial heterogeneity of the influencing factors of transportation carbon emissions. The following suggestions are required to address in order to enhance the quality of the manuscript.
11. Critical literature review is required (reporting other researchers conclusions are not sufficient and critical arguments should include) and the introduction section should more focus on the novelty of the study and industrial/society requirements.
22. Check page 5 lines 168 and 169. Correct the mistake on the reference accordingly.
33. “The number of employees in the transportation industry has no significant effect on transportation carbon emissions in most regions in the south, and in the regions where there is a significant effect, it is mainly a negative effect” (Page 15, lines 479 and 480)
But in the conclusion “(2)….the number of employees in the transportation industry are the main factors influencing…”
The clarification is unclear and difficult to understand. Please provide more clear discussion.
44. The constraints/limitations should be included for every model in the manuscript.
55. The novelty of the study and the industrial significance are unclear. Please specify them in the conclusion.
Author Response
Thank you very much for reading this paper in your busy schedule and for your constructive comments and suggestions on the paper, which have greatly helped us to further improve the paper. We have carefully considered and analyzed your comments, and have revised the paper according to your opinions, and the revision is described as follows.
11.Critical literature review is required (reporting other researchers conclusions are not sufficient and critical arguments should include) and the introduction section should more focus on the novelty of the study and industrial/society requirements.
As you mentioned, the review section of the previous manuscript lacked a critical review of existing studies. This revision summarizes the shortcomings of the existing studies, the novelty of the studies, and the social contributions, and adds our conclusions to the article. The part of the introduction marked in red is the additional content and the details are as follows. The analysis of the spatial and temporal effects of transportation carbon emissions were mostly focused on macroscopic areas, and most scholars analyze them at the national and provincial scales. However, due to the large differences in geographic locations, economic development levels and resource endowments of different cities in China's provincial areas, if the research objects are only limited to the macroscopic levels of national and provincial areas, the relevant policies and recommendations proposed will lack relevance.
22.Check page 5 lines 168 and 169. Correct the mistake on the reference accordingly.
Thank you for your correction, we have corrected the citation errors in the text.
33.“The number of employees in the transportation industry has no significant effect on transportation carbon emissions in most regions in the south, and in the regions where there is a significant effect, it is mainly a negative effect” (Page 15, lines 479 and 480). But in the conclusion “(2)…the number of employees in the transportation industry are the main factors influencing…”. The clarification is unclear and difficult to understand. Please provide more clear discussion.
Thank you for your valuable suggestions, indeed our previous statement may have been ambiguous. The expression " the number of employees in the transportation industry are the main factors influencing…" in the conclusion means that NTP is more important than CR, SR, ET and other indicators among the 12 indicators in the transportation carbon emission impact factor index system constructed by "3.3.1 Indicator system". The NTP is more important. As described in 3.3.4 of the paper: “According to the results of the model, the P<0.05 test was performed on all variables. According to the test results, the variables with a small number of test sample points were screened out, and the variables with a larger number of test sample points were further analyzed. So, we finally selected Intercept, PD, Agdp, NC, ROAD and NTP for further analysis”. That's why the statement in the above conclusion.
- The constraints/limitations should be included for every model in the manuscript.
Your suggestion is very useful to guide our paper. As you mentioned, we have also considered the limitations of the model and elaborated on them in the "Conclusion and discussion (3)" section of the paper, as follows. Although the multi-scale geographically weighted regression makes up for the short-comings of the traditional global regression model, it still has a strong linearity assumption. With the increasing maturity of machine learning and deep learning methods, the linearity assumption of the model should be further relaxed in the future, while exploring the impact of interaction effects among variables on transportation carbon emissions.
55.The novelty of the study and the industrial significance are unclear. Please specify them in the conclusion
Thank you for your suggestions. As you mentioned, our previous manuscript lacked a discussion of the innovative and industrial significance of the study in the conclusion section, which we have added in this revision, as follows. This paper extends the macro-level study of transportation carbon emissions from national and provincial scales to city scales, and on this basis the conclusions about the spatial and temporal variation characteristics of transportation carbon emissions are more reasonable. In addition, this paper uses a multi-scale geographically weighted regression model to analyze the influencing factors of transportation carbon emissions, which can accurately calculate the scale of action of each influencing factor, and the carbon emission reduction strategy of transportation industry based on this can be more targeted.
In addition, we have edited the article extensively for English language and style, so please review it again.
